# Low-Frequency Magnetic Fields (LF-MFs) Inhibit Proliferation by Triggering Apoptosis and Altering Cell Cycle Distribution in Breast Cancer Cells

**DOI:** 10.3390/ijms21082952

**Published:** 2020-04-22

**Authors:** Aoshu Xu, Qian Wang, Tingting Lin

**Affiliations:** 1College of Instrumentation and Electrical Engineering, Jilin University, Changchun 130061, China; xuks15@mails.jlu.edu.cn (A.X.); qianw18@mails.jlu.edu.cn (Q.W.); 2Key Laboratory of Geophysics Exploration Equipment, Ministry of Education of China, Changchun 130061, China

**Keywords:** breast cancer cells, low-frequency magnetic fields, PI3K/AKT pathway, apoptosis, GSK-3β

## Abstract

Breast cancer is a common malignancy threatening women’s health around the world. Despite improved treatments for different subtypes of breast tumors that have been put forward, there still exists a poor therapeutic response and prognosis. Magnetic fields, as a non-invasive therapy, have shown anti-tumor effects in vitro and in vivo; however, the detailed mechanisms involved are still not clear. In this study, we found that in exposure to low-frequency magnetic fields (LF-MFs) with an intensity of 1 mT and frequencies of 50, 125, 200, and 275 Hz, separately, the proliferation of breast cancer cells was inhibited and LF-MF with 200 Hz reached the optimum inhibition effect, on exposure time-dependently. Notably, we found that exposure to LF-MF led to MCF-7 and ZR-75-1 cell apoptosis and cell cycle arrest. Moreover, we also discovered that LF-MF effectively increased the level of reactive oxygen species (ROS), suppressed the PI3K/AKT signaling pathway, and activated glycogen synthase kinase-3β (GSK-3β). We demonstrated that the GSK3β activity contributed to LF-MF-induced cell proliferation inhibition and apoptosis, while the underlying mechanism was associated with the inhibition of PI3K/AKT through increasing the intracellular ROS accumulation. These results indicate that LF-MF with a specific frequency may be an attractive therapy to treat breast cancers.

## 1. Introduction

Breast cancer is a malignant tumor that often occurs in females, accounting for 7–10% of all systemic malignant tumors, which seriously affect the physical and mental health of women [1]. In recent years, the age of onset has tended to be younger, and the tumors themselves are highly heterogeneous with a high mortality rate [2]. Surgery, chemotherapy, endocrine therapy, and radiotherapy are still the main treatment methods for breast cancers. However, as breast cancers consist of a range of tumor subtypes with different clinical characteristics, the prognosis and treatment response of breast cancer patients is different [3]. Therefore, it is of great significance to find a novel, general and more efficient treatment therapy for breast cancers.

Magnetic field (MF) therapy has been put forward as a non-invasive and safe approach to cancer, which has the advantage of a high efficiency and low cost, without inducing infections or forming scars [4]. In recent years, magnetic fields have been reported to have beneficial results in cancers [5,6], peripheral nerve regeneration [7], and osteonecrosis [8]. The frequency, intensity and treatment period of the field are all important parameters that affect the ability of MFs to interact with biological processes [9]. Further research has found that low-frequency magnetic fields (LF-MFs), namely, those with a frequency below 300 Hz, possess a variety of effects such as the regulation of immunity [10] and inflammation [11], suppression of angiogenesis [12], contribution to differentiation [13], and induced apoptosis [14]. Although many biological effects of magnetic fields have been reported, the mechanisms involved still remain unclear. 

Reactive oxygen species (ROS), including superoxide and hydrogen peroxide, are one of the main causes of tumors and play an important role in the process of tumor progression, metastasis and apoptosis [15]. Low-frequency magnetic fields have been demonstrated to significantly increase the intracellular ROS levels in many kinds of cells [16]. The PI3K/AKT pathway is a pivotal signaling pathway, which is closely related to the regulation of cell survival, apoptosis, migration and proliferation [17]. The relationship between ROS and the PI3K/AKT pathway in apoptosis regulation has been confirmed in several studies [18,19]. Glycogen synthase kinase-3β (GSK-3β) is an effector of PI3K/AKT, and is a kind of serine / threonine protein kinase, whose biological function is far beyond the glucose synthesis-regulating enzyme it was initially considered. GSK-3β can phosphorylate many substrates, including metabolism and signal protein, transcription factors of cell structural protein, etc., playing an important role in the process of tumor occurrence and development [20,21]. Baihuan Feng et al. reported that exposure to 50 Hz–0.4 mT MF affected mitochondrial permeability by the ROS-regulating phosphorylation of GSK-3β in human amniotic epithelial cells [22]. 

In the present study, we selected LF-MFs with four different frequencies to study the inhibitory effects on breast cancer cells in vitro. The relationship between intercellular ROS generation and the LF-MF-induced cancer cell inhibition effect was analyzed. In addition, the roles of PI3K/AKT and GSK-3β on the LF-MF-induced apoptosis were investigated. These findings will be helpful for further understanding the molecular mechanisms of MF-induced cancer cell apoptosis, providing a novel therapy and possible therapeutic targets for human breast cancer treatment.

## 2. Results

### 2.1. LF-MF Inhibited the Proliferation of Breast Cancer Cells 

To investigate the toxic effects of the magnetic fields at different frequencies, breast cancer cell lines MCF-7, ZR-75-1, T-47D and MDA-MB-468 were exposed to magnetic fields with an intensity of 1 mT at 50, 125, 200 and 275 Hz for 6, 12, 24, or 36 h, separately (Figure 1a). The MTT assay results (Figure 1b) showed that the LF-MF treatment with the mentioned frequencies resulted in decreased viability with the increased duration of treatment in breast cancer cells. Among them, the optimum suppression effect occurred when the frequency was 200Hz. We used human umbilical vein epithelial cell (HUVEC) cell lines to evaluate the toxicity of LF-MFs in normal cells. The viabilities of HUVECs did not markedly reduce after exposure in LF-MF at the four selected frequencies for 6, 12, 24 or 36 h (Figure 1c). 

### 2.2. LF-MF Induced Breast Cancer Cell Apoptosis

In order to further explore the underlying mechanism of breast cancer cell death induced by magnetic fields, MCF-7, ZR-75-1, T47D, and MDA-MB-468 cancer cells were treated by MF (200 Hz, 1 mT) for 24 h. FITC-Annexin V/PI staining assays were carried out to assess the number of apoptotic cells using flow cytometry. The results showed that the apoptosis rates of four cell lines were enhanced after the MF treatment (Figure 2a). Additionally, the Western blotting analysis of MCF-7 and ZR-75-1 cells showed that the expressions of apoptosis-related proteins were regulated by exposure time-dependently. With the increase in exposure duration, an apparent elevation of cleaved PARP-1, cleaved caspase-3 and Bax, as well as the downregulation of Bcl-2 (Figure 2b) were in accordance with the previous results for the incremental apoptosis rate. 

### 2.3. LF-MF Altered Cell Cycle Distribution in Breast Cancer Cells

To study the mechanisms underlying the anti-proliferation effects of the magnetic fields, we tested whether MF treatment affected the cell cycle distribution of breast cancer cells. MCF-7 and ZR-75-1 cells were treated with the mentioned MF for 6, 12, and 24 h. Propidium iodide (PI) stained-cells analyzed by flow cytometry revealed that there was an accumulation of cells in the G2-M phase (Figure 3a,b). The influence of LF-MF on the expressions of cyclins was detected by Western blot (Figure 3c), while no significant changes are shown in the levels of Cyclin A, D1 and E. Furthermore, the Western blotting results showed that the MF treatment exhibited a time-dependent decrease in the expression level of Cyclin B1, indicating a failure of the transition from the G2 phase to M phase (Figure 3c).

### 2.4. LF-MFs Enhanced the ROS Levels in MCF-7 and ZR75-1 Cells

The levels of ROS in the MCF-7 and ZR-75-1 cells after MF treatment were measured using 2,7-dichlorodi-hydrofluorescein diacetate (DCFH-DA) staining. As shown in Figure 4a, the green fluorescence was markedly brighter than that in the control cells, suggesting that the mean fluorescence intensities were distinctly elevated after LF-MF exposure for 2h. The absorbance read by the multi-mode microplate reader also proved this phenomenon, and this effect could be attenuated by ROS scavenger N-acetyl-l-cysteine (NAC; Figure 4b).

### 2.5. LF-MF Suppressed the Activation of PI3K/AKT Signaling Pathways through ROS Accumulation

The effects of MF on the PI3K/AKT signaling pathway in MCF-7 and ZR-75-1 cells were analyzed using Western blotting. Figure 5a shows that MF exposure downregulated the expressions of p-PI3K and p-AKT in MCF-7 and ZR-75-1 cells, which demonstrated that MF inhibited the activation of the PI3K/AKT signaling pathway. Compared to the group with single exposure to MF, the expressions of p-PI3K and p-AKT were upregulated after MF+NAC treatment, which suggested that NAC reversed the MF-induced inactivation of the PI3K/AKT signaling pathway. These findings indicate that LF-MF suppressed the activation of PI3K/AKT signaling pathways by ROS accumulation. Moreover, as illustrated in Figure 5b,c, the effects of MF on viability and apoptosis were partially reversed by NAC. 

### 2.6. The Activation Function of LF-MF on GSK-3β Was Partially Attenuated by IGF-1 and CHIR-99021

As shown in Figure 6a, the expression levels of p-GSK-3β (Ser 9) in MCF-7 and ZR75-1 cells were remarkably downregulated after LF-MF exposure, implying the activation of GSK-3β, whereas this effect was partly reversed by PI3K/AKT activator IGF-1 and specific GSK-3β inhibitor CHIR-99021. These findings indicate that LF-MF promoted the activation of GSK-3β through the inactivation of PI3K/AKT. Furthermore, the MTT assay revealed that IGF-1 and CHIR-99021 reversed the MF-induced reduction of cell viability in MCF-7 and ZR-75-1 cells (Figure 6b). 

### 2.7. LF-MF Induced MCF-7 and ZR75-1 Cells Apoptosis by Activating GSK-3β

To further explore the role of the GSK-3β activity in LF-MF-induced MCF-7 and ZR-75-1 cell apoptosis, CHIR-99021 was used to inactivate GSK-3β in MCF-7 and ZR-75-1 cells. As shown in Figure 7a, flow cytometry with Annexin V/PI double staining revealed that CHIR-99021 reversed MF-induced increased apoptosis. Figure 7b presented that the expressions of Bax, cleaved-caspase 3 and cleaved-PARP1 were decreased after MF+CHIR-99021 treatment, compared with the single LF-MF treatment. These results suggested that GSK-3β inhibition attenuated the MF-induced MCF-7 and ZR-75-1 cell apoptosis. 

## 3. Discussion

A large number of studies have shown that low-frequency magnetic fields could inhibit cancer cell proliferation and exert the function of inhibiting malignant tumors. However, the mechanism of this technique remains poorly understood. Because of the variable parameters of the LF-MFs and the cell types used in different research, the biological effects are not in complete accord. As most of studies have been carried out using magnetic fields with a specific frequency and field intensity [10,23,24], a few of them verified the influence of different intensities [25]; however, the influence of frequency has rarely been studied. To explore whether the frequency of LF-MF is an important factor to its anti-tumor effects, we choose four frequencies, namely, 50, 125, 200 and 275 Hz. We found that LF-MFs at the four frequencies inhibited the viability of the breast cancer cells, but had no significant effects on the non-cancerous cell lines. Among the frequencies we used, 200Hz is suitable for the optimum inhibition frequency to breast cancer cells. Notably, as shown in Figure 1, the inhibition effects were not enhanced with the increased frequencies, which meant that LF-MF with a specific frequency rather than a higher one had obvious inhibitory effects on breast cancer cells. Therefore, we inferred that the biological effects of LF-MFs might depend on tumor-specific frequencies, while the optimal frequency to different cell types needs further research. 

Some studies showed that LF-MFs have the potential to induce apoptosis in cancer cells [14,25]. Flow cytometry were used to test whether LF-MF exposure affected the apoptotic rate, and we found that the numbers of apoptotic cells in four cell lines increased significantly after 24 h exposure. To further verify these results, we tested alternations in the expression levels of apoptosis-related proteins following 6, 12, and 24 h treatment with LF-MF in MCF-7 and ZR-75-1 cells. We showed that LF-MF induced cell apoptosis by increasing the expressions of the pro-apoptosis proteins Bax, cleaved caspase-3, and cleaved PARP1, and decreasing the expressions of anti-apoptosis protein Bcl-2. LF-MFs were shown to induce cell cycle arrest at the G0-G1 phase or G2-M phase [10,26], which might be caused by different exposure conditions. It is well known that Cyclin D1 is a key protein regulating the G1 phase [27], whereas the inhibition of Cyclin B1 function results in a G2-M phase arrest [28]. In this study, we found that with continuous exposure to 200 Hz-1 mT LF-MF, the cells in the G2-M phase increased, which is associated with the downregulation of Cyclin B1.

The phenomenon that magnetic fields increase the ROS levels in cancer cells has been proven in several studies, however, whether the anti-tumor effect is associated with ROS accumulation remains unclear [29]. In this study, the images of the fluorescence microscopy show that LF-MF obviously increased the ROS accumulation in MCF-7 and ZR-75-1 cells. Pretreatment with NAC, a scavenger of ROS, not only rescued the reduction of cell viability but also mitigated breast cancer cell apoptosis. These results confirmed that the pro-apoptotic effect of LF-MF on breast cancer cells was mediated through excessive ROS generation.

The PI3K/AKT/ GSK3β signaling pathway is one of the important signal transduction pathways, playing an anti-apoptosis and survival-promoting role in cells by activating a variety of downstream effector molecules [30]. Some studies showed that the signaling pathway is closely related to the occurrence and development of a variety of common malignant tumors, as the expressions of p-AKT and p-GSK3β are significantly higher than those in normal tissues within tumors like glioma, ovarian carcinoma, and breast carcinoma [31,32,33,34]. Hongen Lei et al. showed that LF-MF decreased human prostatic adenocarcinoma cell proliferation and the activity of AKT pathway [35]. Lei Zhang et al. found that 1 T moderate intensity static magnetic field inhibited AKT/mTOR in different cell lines [36,37]. In the current study, we also found that LF-MF treatment downregulated the expression of p-PI3K and p-AKT in MCF-7 and ZR-75-1 cells, which means that LF-MF inhibited the PI3K/AKT signaling pathway. However, NAC co-treatment reversed the LF-MF-induced inactivation of the PI3K/AKT signaling pathway, which verified that LF-MF induced MCF-7 and ZR-75-1 cell apoptosis by suppressing the PI3K/AKT signaling pathway through ROS accumulation. Therefore, we confirmed that the enhancement of ROS levels played an important role in LF-MF-induced inactivation of the PI3K/AKT signaling pathway in breast cancer cells.

As is known, AKT substrate GSK3β is negatively modulated by AKT activity, while activated GSK3β-Ser9 (non-phosphorylated state) is reported to regulate cell cycle analysis and apoptosis [38,39,40]. In our research, we also analyzed the effects of LF-MF on the activities of GSK3β in MCF-7 and ZR-75-1 cells. Our results found that LF-MF obviously elevated the GSK3β activity by downregulating p-GSK3β (Ser 9). Pretreatment with IGF-1, PI3K/AKT agonist, or CHIR-99021 (an inhibitor of GSK3β), could recover the expression of p-GSK3β. Moreover, co-treatment with CHIR-99021 reversed LF-MF-induced increases of Bax, cleaved caspase-3, and cleaved PARP1 as well as decreases of Bcl-2 in MCF-7 and ZR-75-1 cells, which signified a reduced apoptosis rate. Then, we can get a conclusion that LF-MF promoted MCF-7 and ZR-75-1 cell apoptosis through the ROS-PI3K/AKT/ GSK3β signaling pathway.

In conclusion, the current findings suggested that although the LF-MFs with the four selected frequencies effectively inhibited the proliferation of breast cancer cells, the best inhibition was obtained at a frequency of 200 Hz. LF-MFs inhibited the AKT signaling network by increasing the levels of ROS, which would lead to apoptosis induction in breast cancer cells. Furthermore, we propose that GSK-3β may be a therapeutic or preventive target for LF-MF induced apoptosis. Specific frequency magnetic fields might be a promising therapy for treating tumors, while further in vivo studies are still needed to validate these findings in current research.

## 4. Materials and Methods 

### 4.1. LF-MF Exposure System

The exposure system was composed of a signal generator, power amplifier, power supply system, and Helmholtz coils (Figure 8a). Helmholtz coils (in a pair) are common devices for generating magnetic fields with a certain uniform space; in order to obtain a larger uniform space to cover different sizes of Petri dishes, we replaced the traditional two coils with three coaxial coils. The outside two coils were designed with 64 turns, while the middle one had 50 turns with the same radius of 130 mm. As a result, the improved Helmholtz coils achieved a 220 × 220 × 116 mm^3^ nominal volume, suitable for the simultaneous exposure of a large number of cell flasks for the parallel studies. The three-dimensional (3D) numerical modeling of the magnetic field produced by the improved Helmholtz coils was accomplished using Multiphysics software (Version 5.3, Comsol, Stockholm, Sweden; Figure 8b–e).

The signal generator was Agilent 33220A 20 MHz Function/Arbitrary Waveform Generator (Agilent Technologies, CA, USA), which could output a sine wave with a frequency range from 1 uHz to 25 MHz. However, the power provided by the generator was not enough to drive the coils to generate a magnetic field with several millitesla. A class A/B audio amplifier system was chosen for the power amplification, which could realize current amplification up to 40 times, while the power supply voltage was ±40 to ±80 V. A ring-shaped transformer was used for the power supply system, converting 220 V AC to 36 V AC, and then filtering by a bridge rectifier board, 36 V DC was generated for driving the amplifier.

The designed Helmholtz coils were placed inside the incubator, and the temperature around the culture medium was monitored with a handheld CTH 6200 thermometer (WIKA Wiegand, Klingenberg, Germany). Measurements were performed at a distance of about 1 cm from the culture medium, and no appreciable increase in temperature was measured during exposure. The unexposed sham samples were placed into another incubator to shield from the magnetic fields.

### 4.2. Reagents

The ROS inhibitor N-acetyl-l-cysteine (NAC) was purchased from Beyotime Institute of Biotechnology. Recombinant Human IGF-I was obtained from Peprotech. CHIR-99021 was purchased from Selleck Chemicals. The primary antibodies for specific detection against GAPDH, Bax, Bcl-2, PARP1, Cyclin B1, PI3K, p-AKT (Ser473), AKT, p-GSK-3β (Ser 9) and GSK-3β were purchased from Cell Signaling Technology (Beverly, MA, USA), while the anti-p-PI3K and anti-cleaved caspase-3 antibodies were purchased from Abcam Technology (Cambridge, MA, USA). HRP conjugated goat anti-rabbit IgG and goat anti-mouse IgG were purchased from Santa Cruz Biotechnology (Santa Cruz, CA, USA).

### 4.3. Cell Culture

The human breast cancer cells MCF-7, ZR75-1, T47D and MDA-MB-468 were obtained from the American Type Culture Collection (ATCC). The cells were cultured in Dulbecco’s modified Eagle medium (DMEM), with 10% fetal bovine serum, 100 U/mL penicillin and 100 U/mL streptomycin and were maintained at 37 °C in a humidified atmosphere with 5% CO_2_. The HUVECs (human umbilical vein epithelial cells) were obtained from Shanghai Institute of Cell Biology, Chinese Academy of Sciences (Shanghai, China) and were cultured according to the provided instructions. Cells in logarithmic growth period were used in the experiments.

### 4.4. Cell Viability Assay

The proliferation of breast cancer cells was determined by MTT assay. Briefly, breast cancer cells were seeded into 96-well microplates, with 1 × 10^4^ cells per well and were cultured for 24h. Thereafter, the cells were exposed to magnetic fields for indicated periods, and then the MTT solution was added into each well. The ratio of absorbance value at 570nm of treated cells to the control cells was thought to represent the cell survival rate.

### 4.5. Apoptosis Assays

Cellular apoptosis was assessed by flow cytometry analysis following the manufacturer’s instructions of an Annexin V-FITC Apoptosis Detection Kit (Becton Dickinson, San Diego, CA, USA). For a brief description, cells were seeded into a six-well plate at a density of 2 × 10^5^. After specific treatment within each group, at least 10,000 cells were harvested and then counted and analyzed by a flow cytometer (Beckman Coulter, Brea, CA, USA).

### 4.6. Cell Cycle Analysis

A cell cycle analysis kit was bought from Yeasen. Breast cancer cells were exposed to LF-MF for 6, 12, and 24 h separately, and then collected. After washing with PBS, the cells were fixed in 70% ethanol overnight at −20 °C and were treated with 0.5 mL of staining buffer containing 10 uL PI and 10 uL RNase in the dark for 30 min. The stained cells were subsequently analyzed using a flow cytometer (Beckman Coulter, Brea, CA, USA).

### 4.7. Western Blotting

After the indicated treatment, MCF7 and ZR75-1 cells were collected and total proteins were extracted using a RIPA buffer (Beyotime Biotechnology, Shanghai, China). A BCA Protein Assay Kit was used to calculate the concentrations of the protein samples following the instruction. After quantification, equal amount of protein samples were loaded and separated on SDS-PAGE gel. Then, the proteins in the gel were transferred to PVDF membranes (0.22 μM, Millipore, MA, USA) and subsequently blocked by 5% non-fat dried milk in TBS for 1h at room temperature. After that, the membranes were incubated with the corresponding primary antibodies overnight at 4 °C. Finally, the membranes were incubated with a goat anti-rabbit or anti-mouse IgG (H + L) secondary antibody at room temperature for 1h. The signals were detected by enhanced chemiluminescence and the images were acquired by a Tanon-4600SF Imaging System (Tanon, Shanghai, China).

### 4.8. Measurement of Intracellular ROS Level

The average levels of intracellular ROS in MCF7 and ZR75-1 cells were detected using 2,7-dichlorodi-hydrofluorescein diacetate (DCFH-DA) by a ROS Assay Kit from Beyotime. MCF-7 and ZR-75-1 breast cancer cells (1 × 10^5^ cells/well) were seeded into six-well plates and were cultured for 24 h. After exposure to the mentioned magnetic field for 3 h, with or without pre-treatment with NAC (5 mM) for 1h, the cells were stained with 10 uM DCFH-DA at 37 °C for 30 min in the dark and washed with PBS for three times. After that, the intracellular ROS levels of cells were captured by a fluorescence microscope (Olympus IX53/DP80, Tokyo, Japan) for fluorescent photo and were analyzed by a multi-mode microplate reader (Biotek, Synergy H1, Winooski, VT, USA) for the fluorescence intensity.

### 4.9. Statistical Analysis

All of the results were obtained from at least three independent experiments and are presented as the means ± standard deviation (SD). One-way analysis of variance (ANOVA) was carried out for statistical comparisons, and *p*-values below 0.05 were considered to represent statistical significance.

## Figures and Tables

**Figure 1 ijms-21-02952-f001:**
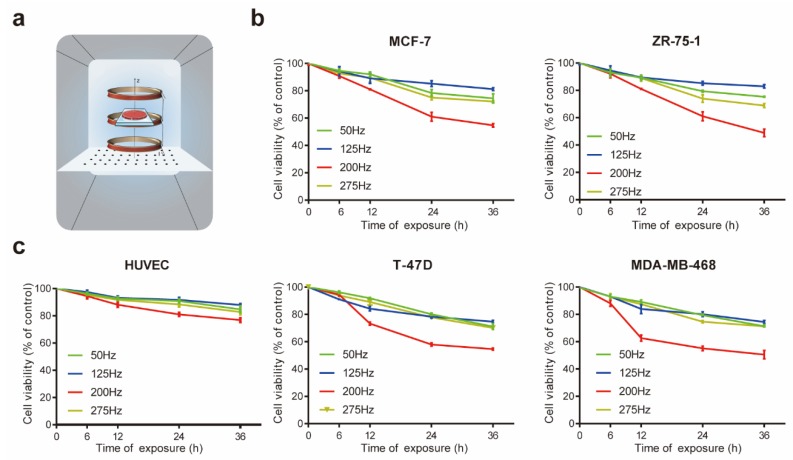
Low frequency magnetic fields inhibited the viabilities of breast cancer cells. (**a**) A three coil Helmholtz system is settled in the cell incubator to provide a magnetic field (MF) condition for cell culture. (**b**) and (**c**) 3-(4,5-dimethyl-2-thiazolyl)-2,5-diphenyl-2-H-tetrazolium bromide (MTT)assay showed that low frequency MF inhibited the proliferation of MCF-7, ZR-75-1, T-47D and MDA-MB-468 cells in a time-dependent manner, but had limited toxicity in HUVECs. Data were presented as mean ± standard deviation (SD); *n* = 3.

**Figure 2 ijms-21-02952-f002:**
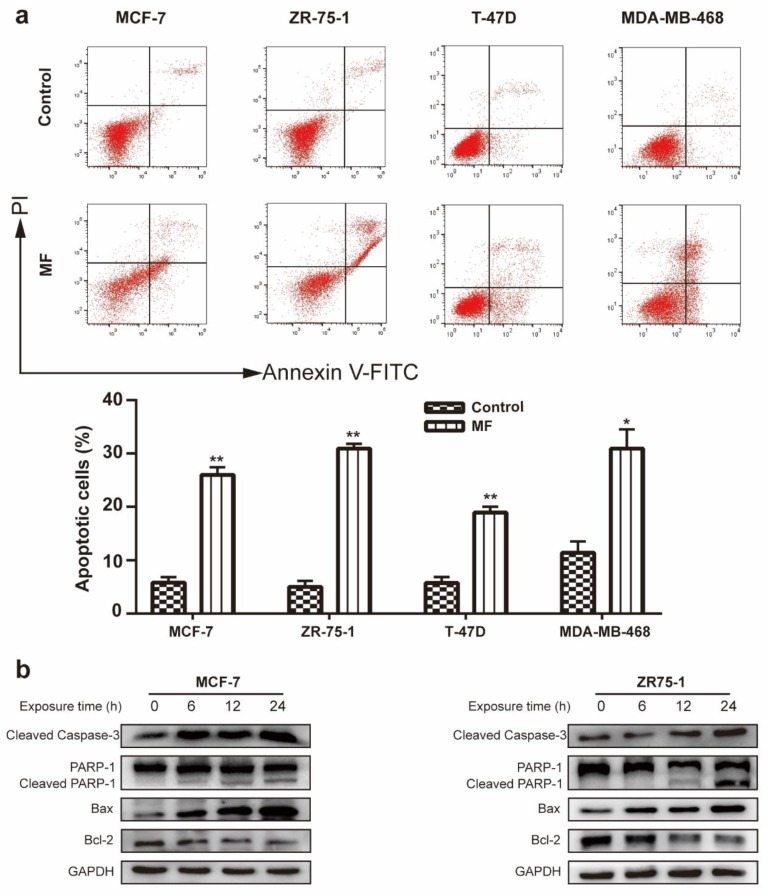
Effects of MF on breast cancer cell apoptosis. (**a**) Flow cytometry images (up) showing the expression levels of Annexin V- and PI-labeled MCF-7, ZR-75-1, T47D, and MDA-MB-468 cells following a 24 h-treatment with or without MF (200 Hz, 1 mT). Histograms (down) illustrating the number and distribution of apoptotic cells in the total cell population. (**b**) Western blotting analysis of the expression levels of cleaved caspase-3, PARP1, Bax, and Bcl-2 in both MCF-7 and ZR-75-1 cells following exposure to MF for each designated time. In addition, GAPDH was used as a reliable internal control. Data are presented as the mean ± standard deviation (SD); *n* = 3; * *p*< 0.05, ** *p* < 0.01.

**Figure 3 ijms-21-02952-f003:**
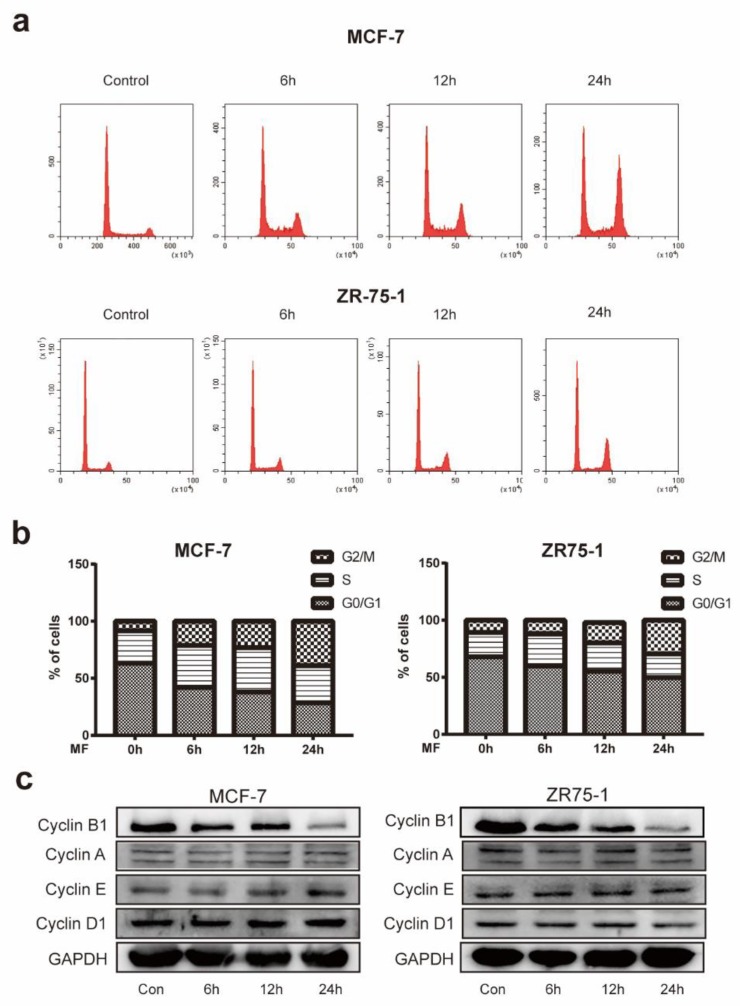
Effects of MF on cell cycle distribution. (**a**) Representative flow cytometry results evaluating the numbers of MCF-7 and ZR-75-1 cells during the G0-G1/S/G2-M phase in both the control and experimental groups treated by MF. (**b**) Histograms showing the percentage of MCF-7 and ZR-75-1 cells in G0-G1, S or G2-M phase in the control and treatment groups with various exposure times to MF. (**c**) Representative western blot results showing the expression level of Cyclin B1, Cyclin A, Cyclin E and Cyclin D1 in MCF-7 and ZR-75-1 cells following exposure in MF for the designated time. In addition, GAPDH was used as a reliable internal control. Data are presented as the mean ± standard deviation (SD); *n* = 3; * *p*< 0.05, ** *p* < 0.01.

**Figure 4 ijms-21-02952-f004:**
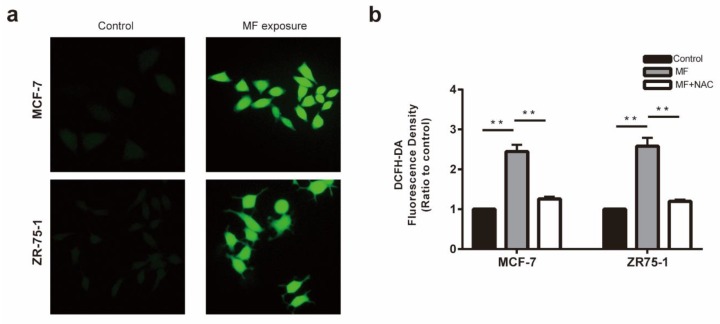
MF enhanced the ROS levels and inhibited the activities of the PI3K/AKT signaling pathways in MCF-7 and ZR-75-1 cells. (**a**) The representative images under a fluorescence microscope (40×) showed that excessive ROS was produced after 2 h of exposure to MF. (**b**) Statistics analysis of the fluorescence intensity in the cells stained with DCFH-DA, which was read from the multi-mode microplate reader, showed the ROS level increased significantly in MCF-7 and ZR-75-1 cells exposed to MF. Data are presented as the mean ± standard deviation (SD); *n* = 3; ** *p*< 0.01.

**Figure 5 ijms-21-02952-f005:**
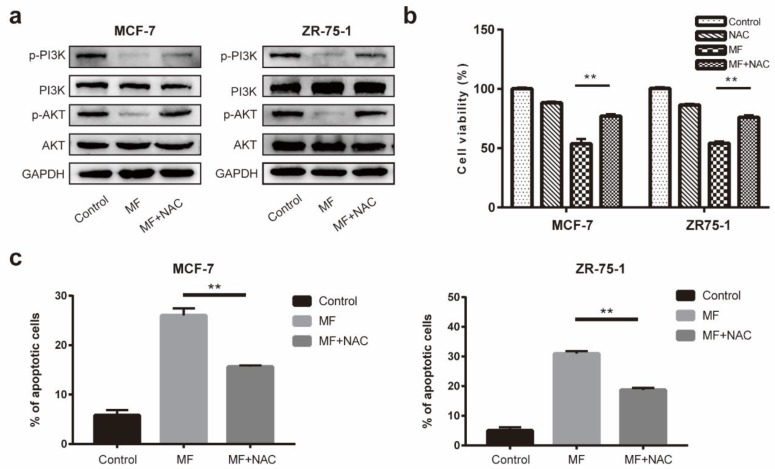
MF inhibited the activities of the PI3K/AKT signaling pathways through ROS accumulation in MCF-7 and ZR-75-1 cells. (**a**) Western blotting was performed to analyze the expression levels of PI3K, p-PI3K, AKT and p-AKT in MCF-7 and ZR-75-1 cells after MF (200 Hz, 1 mT, 24 h) and MF+NAC (N-acetyl-l-cysteine) treatment. (**b**) The viabilities of MCF-7 and ZR-75-1 cells after the MF and MF+NAC treatments were detected using MTT assay. (**c**) Flow cytometry analysis with FITC-PI/Annexin V staining showed pretreatment with NAC rescued part of the MF-induced MCF-7 and ZR-75-1 cell apoptosis. Data are presented as the mean ± standard deviation (SD); *n* = 3; ** *p* < 0.01.

**Figure 6 ijms-21-02952-f006:**
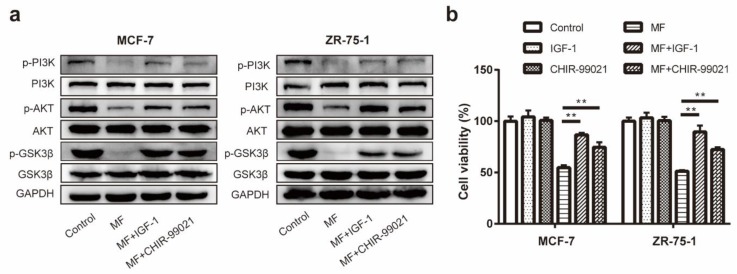
The activation function of LF-MF on GSK-3β was partially attenuated by IGF-1 and CHIR-99021. (**a**) Western blotting was performed to confirm that activating PI3K/AKT (IGF-1) or inactivating GSK-3β (CHIR-99021) suppressed MF-induced p-PI3K and p-AKT and p-GSK-3β decrease. (**b**) MTT assay showing that GSK-3β inactivation reversed the reduced viability caused by MF exposure. *n* = 3; ** *p* < 0.01.

**Figure 7 ijms-21-02952-f007:**
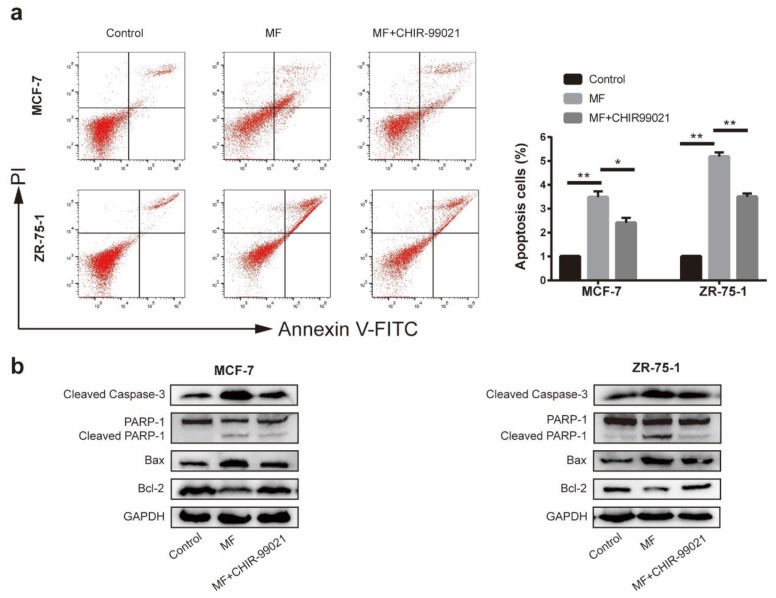
LF-MF induced MCF-7 and ZR75-1 cells apoptosis by activating GSK-3β. (**a**) (left) Flow cytometry analysis with Annexin V/PI staining showing that inactivating GSK-3β rescued MF-induced apoptosis. (right) Histograms show the mean percentage of cells with apoptosis. (**b**) Western blotting was performed to analyze the expressions of cleaved caspase-3, PARP1, Bax, and Bcl-2 in MCF-7 and ZR-75-1 cells after MF treatment and/or CHIR-99021. Data are presented as the mean ± standard deviation (SD); *n* = 3; * *p*< 0.05, ** *p* < 0.01.

**Figure 8 ijms-21-02952-f008:**
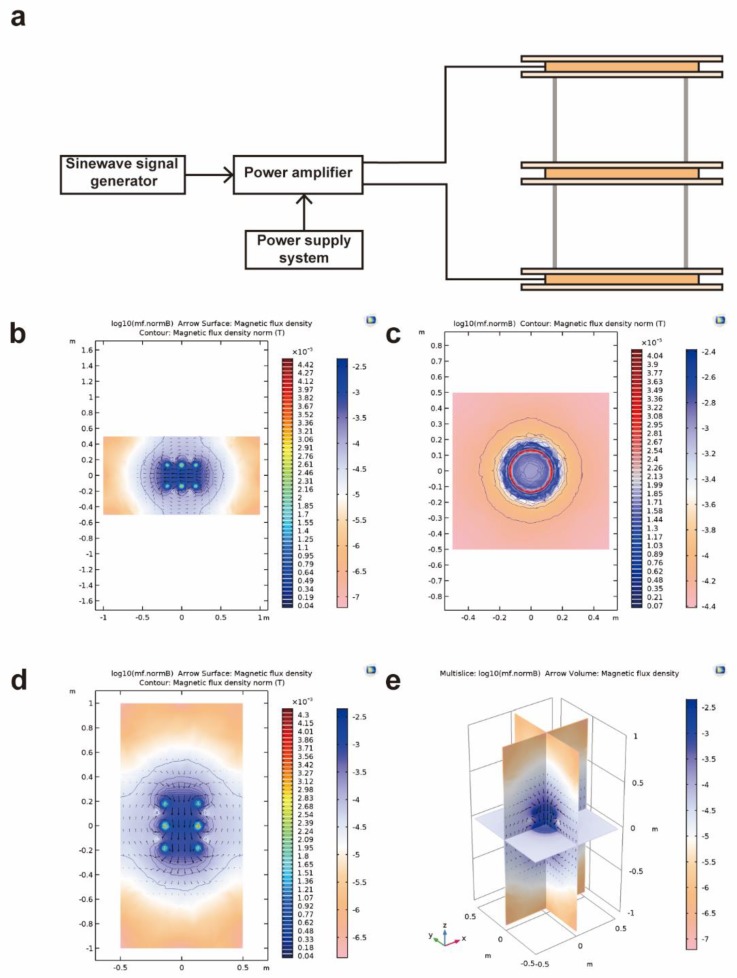
Schematic diagram of the MF exposure system and simulation of MFs produced by the Helmholtz coils. (**a**) The MF exposure system consists of a waveform generator, power amplifier, power supply, and improved Helmholtz coils. (**b**) X-Z plane of 2D magnetic field simulation of three coils at y = 0. The color bar represents the relative value of the magnetic field using a log base of 10. The arrow shows the direction of the magnetic field and the contour represents isoline. (**c**) X-Y plane of 2D magnetic field simulation of three coils at z = 0. (**d**) Y-Z plane of 2D magnetic field simulation of three coils at x = 0. (**e**) Three-dimensional simulation of the Helmholtz coils.

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
