# Peer review of "Low-Frequency Magnetic Fields (LF-MFs) Inhibit Proliferation by Triggering Apoptosis and Altering Cell Cycle Distribution in Breast Cancer Cells"

_ijms, 2020, doi:10.3390/ijms21082952_

Round 1

Reviewer 1 Report

the authors of this research paper tried to understand how Low-frequency magnetic fields (LF-MFs) at different frequencies induce cell death (Apoptosis). They showed increased ROS, suppressed the PI3K/AKT signaling pathway and activating GSK-3β leading to induction of cell death.

1. Discuss how cell proliferation was inhibited more at 200 Hz than 250Hz.

2. Can you also measure effect of LF-MFs on other cyclins too like A, D and E. 

Reviewer 2 Report

LF-MF is thought to be non-invasive and non-ionizing treatment and also possess non-thermal effects on cells and tissues. Thus the possible effects of LF-MF on human health have been attractively explored. The subject discussed in the manuscript is  interesting and worth continuing.

In my opinion the manuscript is written clearly and comprehensibly. Each part of the work is described in details. There are some little editorial errors, for example

section 4.4 -  1×104 cells instead 1 x 104 cells

section 4.5 -  density of 2 × 105  instead density of 2 x 105   

I'm wondering why the authors chose only MCF-7 and ZR-75-1 cells to apoptosis experiment? Do they assume that in case of other lines (T-47D and MDA-MB-468) apoptosis through the indicated path also occurs? 

In my opinion, the experiments with the non-cancerous cells as a control should be also presented.

In discussion, I suggest the Authors take into account interesting publication of Ren et al. 2017 (https://doi.org/10.1038/s41598-017-00913-2) on the investigation of LF-MF-induced apoptosis mechanism in lung cancer.
